

# Significance of hub genes and immune cell infiltration identified by bioinformatics analysis in pelvic organ prolapse

Ying Zhao, Zhijun Xia, Te Lin and Yitong Yin

Department of Obstetrics and Gynecology, Pelvic Floor Disease Diagnosis and Treatment Center, Shengjing Hospital of China Medical University, Shenyang, China

## ABSTRACT

**Objective:** Pelvic organ prolapse (POP) refers to the decline of pelvic organ position and dysfunction caused by weak pelvic floor support. The aim of the present study was to screen the hub genes and immune cell infiltration related to POP disease.
**Methods:** Microarray data of 34 POP tissues in the GSE12852 gene expression dataset were used as research objects. Weighted gene co-expression network analysis (WGCNA) was performed to elucidate the hub module and hub genes related to POP occurrence. Gene function annotation was performed using the DAVID tool. Differential analysis based on the GSE12852 dataset was carried out to explore the expression of the selected hub genes in POP and non-POP tissues, and RT-qPCR was used to validate the results. The differential immune cell infiltration between POP and non-POP tissues was investigated using the CIBERSORT algorithm.
**Results:** WGCNA revealed the module that possessed the highest correlation with POP occurrence. Functional annotation indicated that the genes in this module were mainly involved in immunity. *ZNF331*, *THBS1*, *IFRD1*, *FLJ20533*, *CXCR4*, *GEM*, *SOD2*, and *SAT* were identified as the hub genes. Differential analysis and RT-qPCR demonstrated that the selected hub genes were overexpressed in POP tissues as compared with non-POP tissues. The CIBERSORT algorithm was employed to evaluate the infiltration of 22 immune cell types in POP tissues and non-POP tissues. We found greater infiltration of activated mast cells and neutrophils in POP tissues than non-POP tissues, while the infiltration of resting mast cells was lower in POP tissues. Moreover, we investigated the relationship between the type of immune cell infiltration and hub genes by Pearson correlation analysis. The results indicate that activated mast cells and neutrophils had a positive correlation with the hub genes, while resting mast cells had a negative correlation with the hub genes.
**Conclusions:** Our research identified eight hub genes and the infiltration of three types of immune cells related to POP occurrence. These hub genes may participate in the pathogenesis of POP through the immune system, giving them a certain diagnostic and therapeutic value.

Corresponding author
Zhijun Xia, xiazj@sj-hospital.org

## INTRODUCTION

Pelvic organ prolapse (POP) is caused by dysfunction of the pelvic floor supporting structures and affects the quality of life of many women (*Nygaard et al., 2014*). The prevalence of POP is expected to reach 46% by 2050 (*Wu et al., 2009*). Currently, only approximately 20% of women have POP surgery during their lifetime (*Haya et al., 2018*); however, the reoperation rate is high (*Friedman, Eslick & Dietz, 2018*), which causes a huge economic burden. POP is a complex, multifaceted disease resulting from the interaction between environmental and genetic factors. Pregnancy, vaginal pull-up, time of delivery, age, and obesity have been identified as risk factors (*Hendrix et al., 2002*); nevertheless, the molecular mechanism of POP remains unclear, and there is a lack of suitable prevention and treatment measures in clinical practice. Therefore, it is imperative to explore the molecular mechanism of POP occurrence for the benefit for both women and society.

Owing to the development of gene chips and new generation sequencing technology, bioinformatics analysis plays an increasingly important role in biomedical research. Weighted gene co-expression network analysis (WGCNA) can identify gene module characteristics and hub genes to connect the gene modules and sample characteristics (*Wang et al., 2020b*). The CIBERSORT algorithm can be employed to evaluate immune cell infiltration in tissues based on gene expression datasets. Recently, abundant researches have used this algorithm to explore the function of immune cells in diseases (*Lin et al., 2020*), breast ductal and lobular carcinoma (*Zhang et al., 2019*), osteoarthritis (*Cai et al., 2020*), and high-grade serous ovarian cancer (*Liu et al., 2020*). Our study found hub genes and immune cells highly related to POP occurrence by analyzing POP expression spectrum data in public databases using WGCNA and the CIBERSORT algorithm, providing novel ideas and methods for the treatment of POP. Finally, the differential immune cells was investigated between POP and non-POP tissues.

## MATERIALS AND METHODS

### Obtaining the training and validation POP datasets

The workflow diagram is presented in Fig. 1. The training dataset was obtained from the Gene Expression Omnibus database (GEO, https://www.ncbi.nlm.nih.gov/) in NCBI based on the keywords "pelvic organ prolapse" and "gene expression profiles" and "*Homo sapiens*" (*Brizzolara, Killeen & Urschitz, 2009*). In addition, we selected a dataset with a sample size greater than 30, which is the minimum sample size required to construct a WGCNA network (POP > 15, non-POP > 15); the GSE12852 dataset, containing 16 POP and 18 non-POP patients, was the only dataset to meet this condition. The gene expression profiles and corresponding clinical information including age, menopausal status, race, and prolapse stage were acquired from the GSE12852 dataset. Subsequently, log-scale robust multi-array analysis was used to perform background correction and normalization of the datasets. A total of 12 POP patients enrolled at the Shengjing Hospital of China Medical University from 2017 to 2018 were selected as the validation dataset. The Ethics Committee of Shengjing Hospital of China Medical University approved the
Downloading data of GSE12852 from GEO database as training dataset

Remove outlier sample of GSE12852 dataset by "flash cluster" package of R software

Construct co-expression network and screen hub module

Construct PPI network using genes in hub module

GO and KEGG enrichment analysis for hub module

Screen hub genes and validate hub genes using the validation dataset

CIBERSORT algorithm characterized the proportion of 22 immune cells in samples from GSE12852 dataset

Investigation the differential immune cells infiltration between POP tissues and non-POP tissues

**Figure 1 Workflow diagram.** Data collection, analysis and validation.
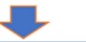

study protocol in accordance with the guidelines of the Declaration of Helsinki (No. 2018Ps68K), and our study was exempt from consent. Prolapse tissues were selected as the experimental group and non-prolapse tissues were selected as the control group. All patients were married, without estrogen-related disorders, and had received no hormone treatment within the previous 3 months. Basic information of the patients in the validation dataset is presented in Table S1.

## Co-expression network construction based on the training dataset

We used the "WGCNA" package of the R software to establish the network based on the 34 samples in the training dataset according to a previously described method (*Chen et al., 2018b*; *Zhou et al., 2018*). Firstly, POP and normal samples were analyzed using the "flash cluster" package of the R software to remove outlier samples. The correlation matrix of gene co-expression consists of the correlation coefficient between two genes. The average linkage matrix and Pearson correlation method were performed to construct the correlation matrix between genes. Subsequently, the formula $amn = |cmn|^{\beta}$ was used to convert the correlation matrix to the adjacency matrix ($amn$ represents the correlation coefficient between gene $m$ and gene $n$; $cmn$ represents the connection coefficient between gene $m$ and gene $n$; and $\beta$ is a soft threshold that can strengthen strong links between genes

and weaken weak links). Finally, we converted the adjacency matrix to the topological overlap matrix based on an appropriate soft threshold, and placed the similar genes into the same module.

## Identification of hub gene modules and genes

Similar genes were placed into the same module based on the WGCNA co-expression network. The main purpose of our research was to combine clinical information (POP or non-POP samples) with gene modules to analyze gene significance (GS) and modular membership (MM). MM means the correlation between gene expression profile level and module eigengenes (ME). ME are considered the correlation between modules and clinical information. GS represents the degree of correlation between gene expression profiles and clinical information. The average value of all gene GS in the module represents the module significance (MS). We defined the correlation between gene and disease as GS, and obtained the correlation between this module and disease as MS. The hub genes characterized by a high MM and a high GS are described as having the closest relationship with disease. In our research, the module with the highest MS was selected as the hub module. The genes in the hub module with $|MM| > 0.8$ and $|GS| > 0.47$ were considered hub genes.

## Functional enrichment analysis

We used the online GO enrichment analysis and KEGG pathway analysis tools on the DAVID website to annotate the genes in the module identified by WGCNA, and attempted to elucidate the enrichment pathway and functions of the target genes. The Human Genome U133 Plus 2 Array was used as the background data (The FDR < 0.01).

## Immune cell infiltration in POP tissues

CIBERSORT is an algorithm to characterize the proportion of 22 immune cells (Table S2) in tissues using 547 barcode gene expression values. The CIBERSORT algorithm was employed to elucidate the proportion of 22 immune cells in POP tissues. The samples with $p$ value < 0.05 were significant (*Chen et al., 2018a*). Pearson correlation analysis was implemented to obtain the related coefficient between the 22 immune cells. Then, we investigated the differential immune cell infiltration between POP and non-POP tissues. Finally, we calculated the related coefficient between infiltration of different immune cells and hub genes by Pearson correlation analysis.

## RNA extraction and quantitative real-time PCR based on the validation dataset

TRIzol® (one mL) was used to isolate total RNA from POP tissues (200 mg) in the validation dataset, and reverse transcriptase from the avian myeloblastoma virus and random primers were used to create complementary DNA (cDNA) according to the instructions from TAKARA. SYBR Premix Ex Taq II (Takara, Shiga, Japan) was used to amplify the cDNA. According to the samples from three independent experiments, the $2^{-\Delta\Delta CT}$ value was used to analyze the data. Primers for the genes are displayed in Table S3.
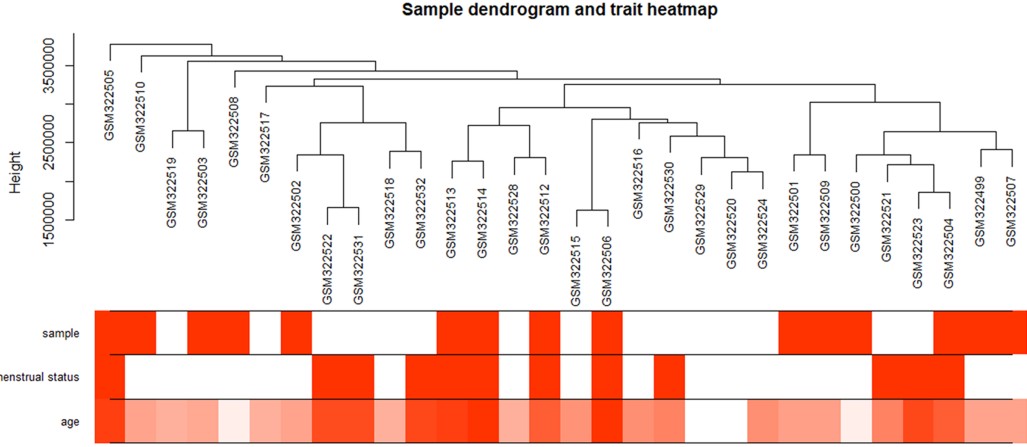

**Figure 2 Clustering dendrogram of 34 POP samples in GSE12852.** POP samples were assigned as 1; non-POP samples were assigned as 0. Pre-menopause status samples were assigned as 0; menopause status samples were assigned as 1. Red color intensity is proportional to POP samples, pre-menopause status, and higher age.

## Statistical analysis

The GraphPad Prism 7.0 and R 3.6.1 software were used for statistical analysis and image generation. A *t*-test was employed to analyze the differences between two groups. The $p < 0.05$ was statistically significant.

## RESULTS

### Construction of WGCNA and identification of hub modules

The "flash cluster" software package in R was used to cluster the 34 samples in the GSE12852 dataset to remove the outlier samples; four outlier samples were removed (Fig. 2). There were few differences among the remaining samples, which is conducive to the accuracy of the results. The correlation matrix between genes was calculated according to the average linkage matrix and Pearson correlation method. The formula $amn = |cmn|^\beta$ ($\beta = 7$) was used to transform the correlation matrix into the adjacency matrix. In order to better construct a scale-free network distribution, the "picks of threshold" function of "WGCNA" package calculated the value of parameter $\beta$. In POP and non-POP samples, 1–20 thresholds were selected and the correlation coefficient, mean connectivity, and average correlation degree between $\log(k)$ and $\log(P(k))$ were calculated for each threshold. At this time, the average network connectivity corresponding to the threshold is close to zero, indicating that the network connectivity is very low, which is similar to the scale-free network (Fig. 3). According to the corresponding steps of WGCNA, the gene network was built after the hierarchical clustering tree. In this experiment, the dynamic pruning tree method merged with similar genes into one gene module. The minimum number of genes in a module was 30, and 11 modules were obtained (Fig. 4A). According to the thermogram of correlation between modules and clinical information, the highest correlation existed in the blue module and sample type, with a correlation coefficient of 0.47 (Fig. 4B; $p = 0.009$). Therefore, the blue module was

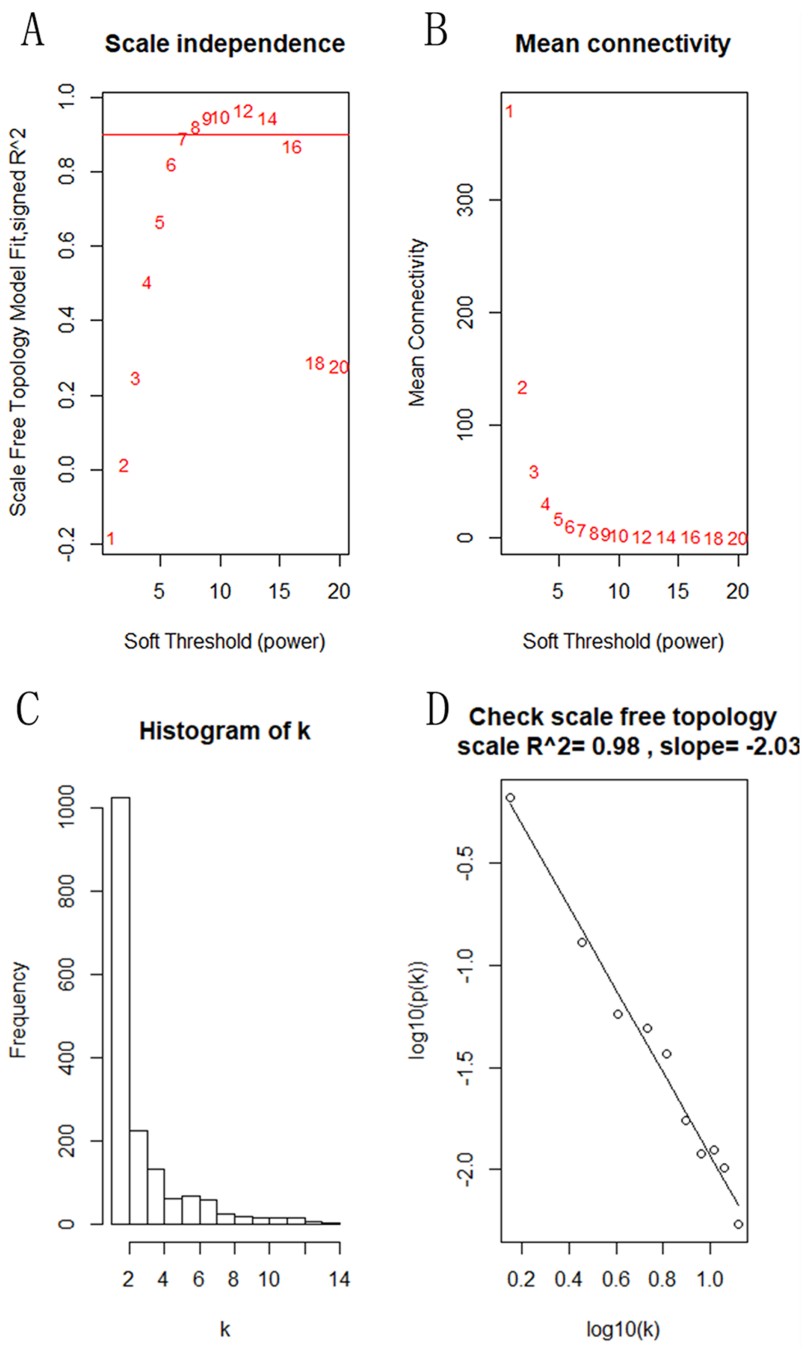

**Figure 3 Identification of the soft threshold in the scale-free network.** (A) Different soft-threshold and corresponding scale free topology model. (B) Different soft-thresholding powers and corresponding mean connectivity. (C) The distribution of the connectedness by histogram. (D) Inspection of the scale free topology.

considered the most study-worthy. Protein–protein interactions (PPIs) are widely involved in the process of vital movement. http://omnipathdb.org/ was used to construct the PPI network for genes in the blue module (Fig. 5) (*Türei, Korcsmáros & Saez-Rodriguez, 2016*).

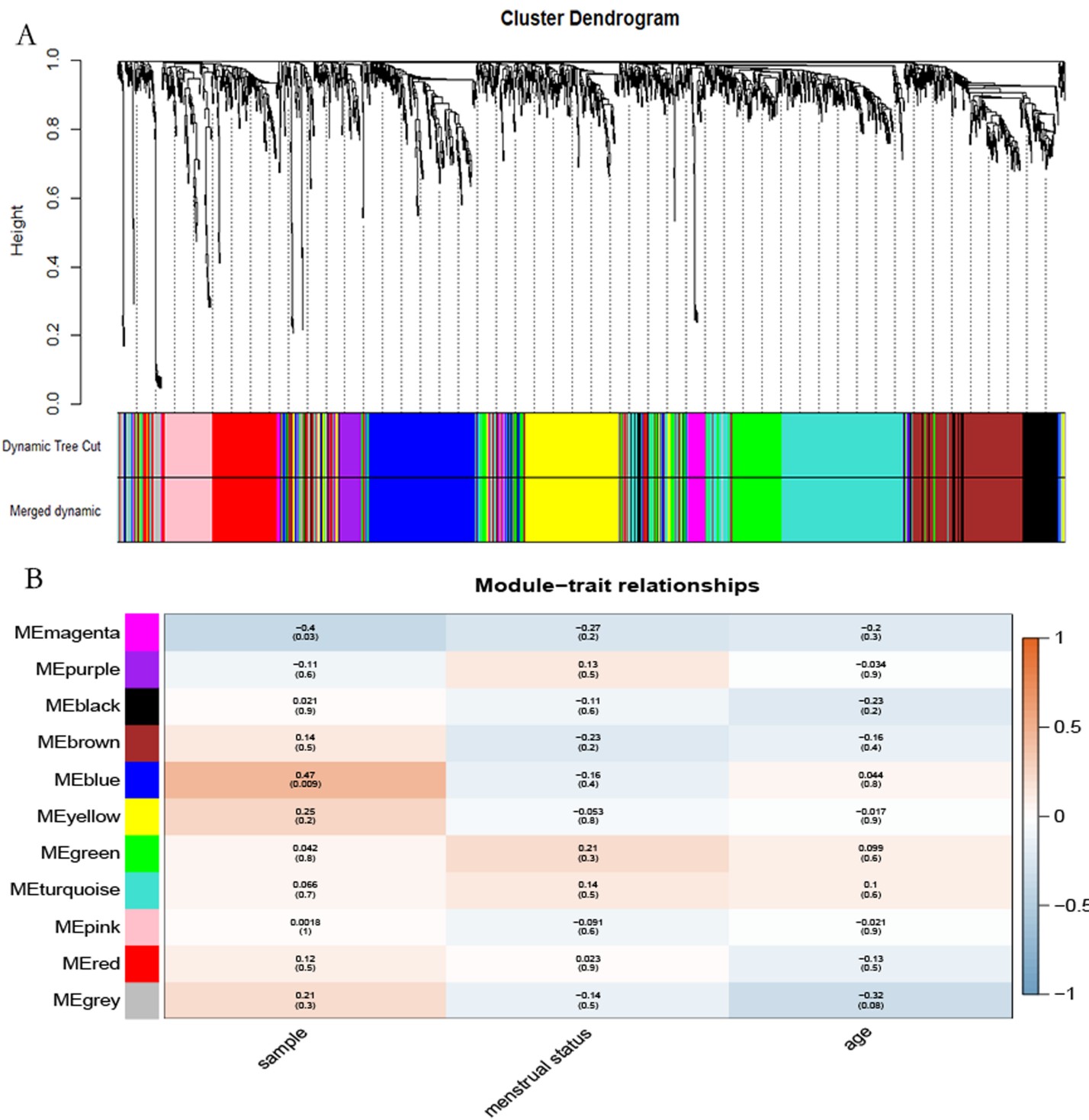

**Figure 4 Identification of the hub module.** (A) Dendrogram showed the genes with similar function classed into the same module according to the dissimilarity measure. (B) Heatmap revealed the correlation between modular eigengenes (ME) and clinical information of POP.

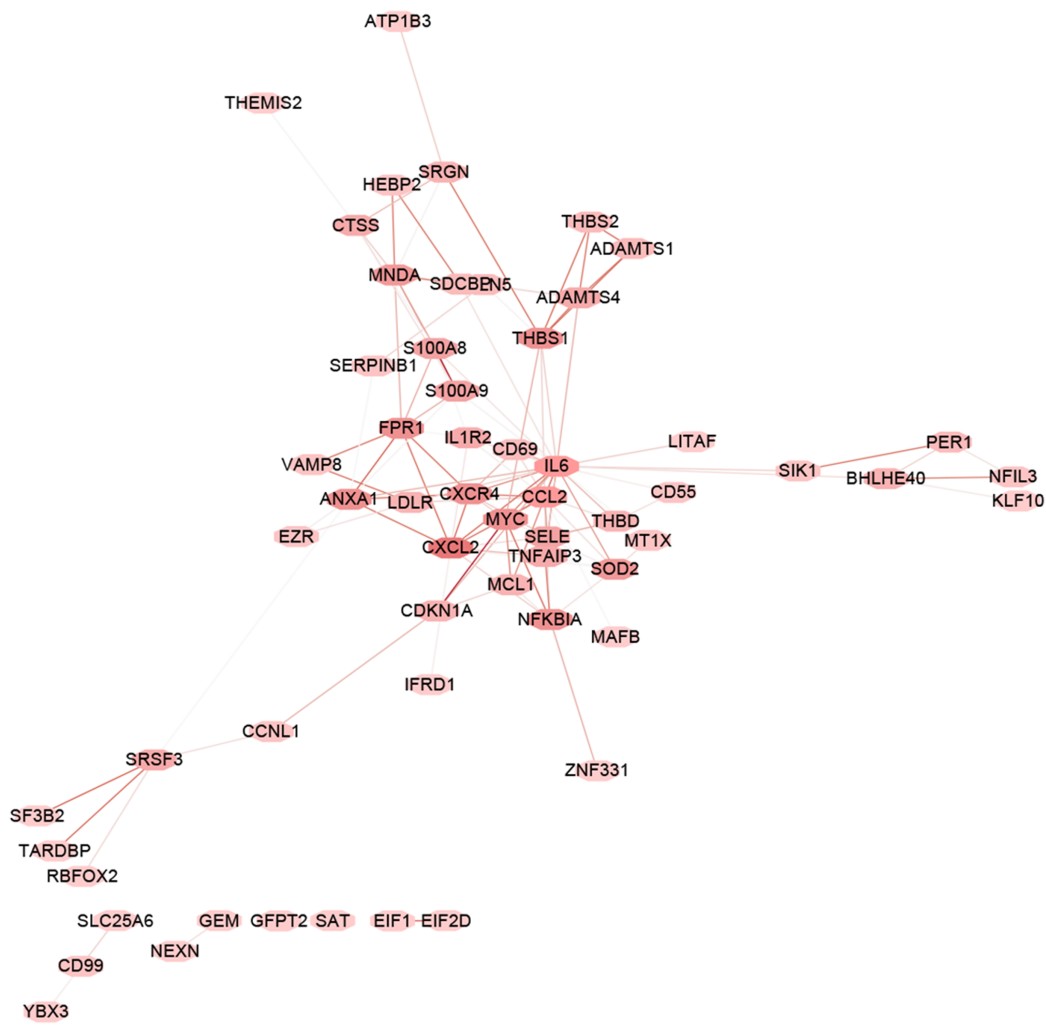

**Figure 5  PPI network analysis.** PPI network analysis of the genes in the blue module based on the OmniPath online database. Nodes represent the hub genes in the blue module identified by WGCNA. Lines represent interactions between hub genes.

## Functional enrichment analysis

To further study the biological function of the genes in the blue module, GO and KEGG enrichment analyses were used. GO enrichment analysis showed that these genes mainly participated in the immune regulation (Fig. 6A; Table S4). KEGG enrichment analysis identified that these genes participated in the regulation of immune-related pathways, such as the IL-17 and TNF signaling pathways (Fig. 6B; Table S5).

## Identification of the hub genes

According to the screening criteria |MM| > 0.8 and |GS| > 0.47, eight genes (*ZNF331, THBS1, TMEM70, CXCR4, GEM, SOD2,* and *SAT*) were identified in the blue module as hub genes (Fig. 7A). We found that the hub genes were highly expressed in POP tissues as compared with non-POP tissues by differential analysis based on the GSE12852 dataset (Fig. 7B). From the heatmap, it can be seen that hub genes are overexpressed in POP

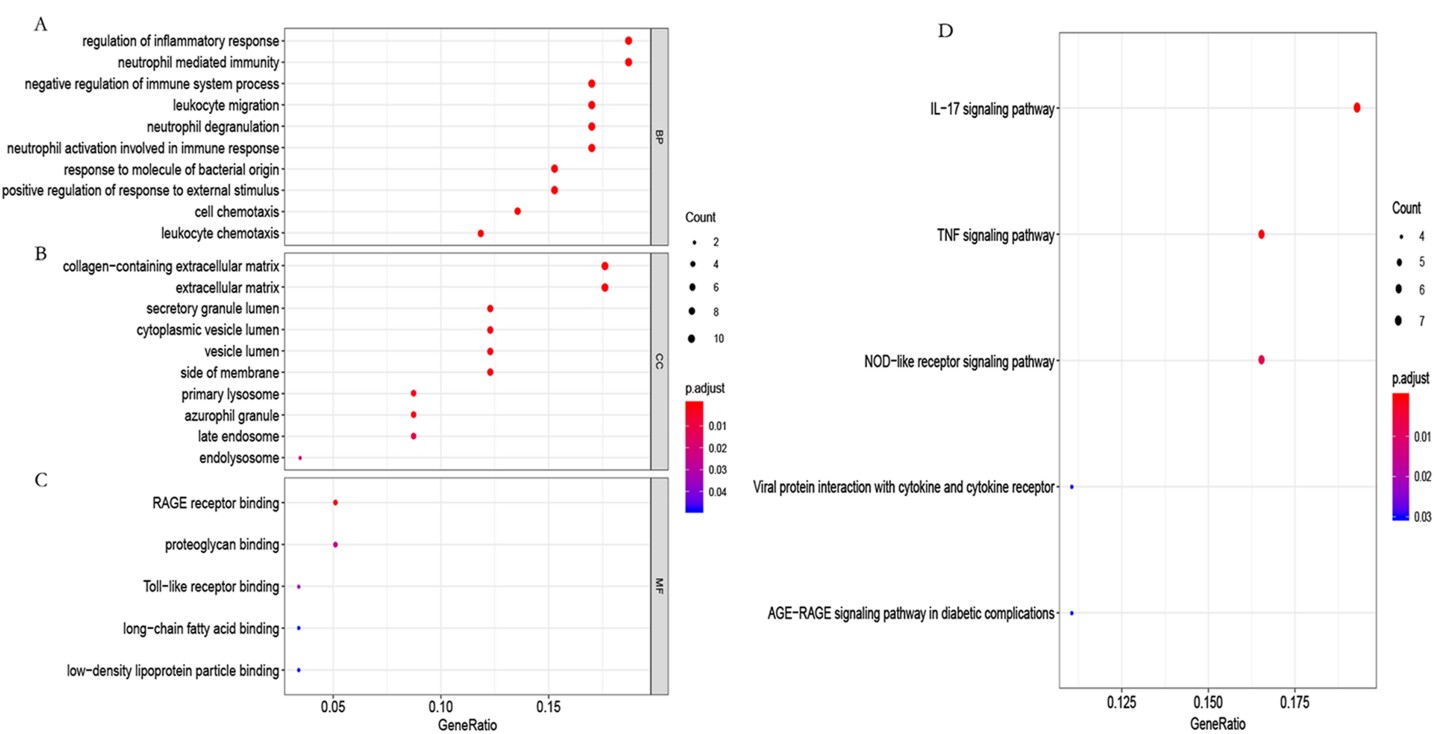

**Figure 6  GO functional and KEGG pathway enrichment analysis.** (A) Biological process (BP), (B) cellular component (CC) and (C) molecular function (MF) of GO analyzed by the "org.Hs.eg.db" package in the R software. (D) KEGG pathway enrichment analysis using the "org.Hs.eg.db" package in the R software.                               

tissues (Fig. 7C). In addition, the correlation coefficient between the hub genes was calculated, which showed that there was a strong co-expression relationship between these genes (Fig. 7D).

## Validation of the hub genes

To verify the accuracy of the prediction results, RT-qPCR was used to detect the expression of the hub genes in 12 pairs of POP and non-POP tissues. The results show that the hub genes were overexpressed in POP tissues, which is consistent with the prediction results (Figs. 8A–8H).

## Immune cell infiltration analysis

The CIBERSORT algorithm was employed to select samples with a output $p < 0.05$. A total of 12 samples including four non-POP and eight POP tissues were obtained. A bar plot was generated to show the proportion of 22 immune cells in the 12 samples (Fig. 9A). We found that macrophages account for the largest proportion among the immune cells in the samples. Figure 9B indicates that M1 macrophages had the strongest positive correlation with resting mast cells (correlation coefficient, 0.58), whereas resting mast cells had the strongest negative correlation with activated mast cells (correlation coefficient, 0.83). We found a higher infiltration of activated mast cells and neutrophils in POP tissues than in non-POP tissues, while the infiltration of resting mast cells was lower in POP tissues (Fig. 10; $p < 0.05$). Finally, Pearson correlation analysis was used to calculate the

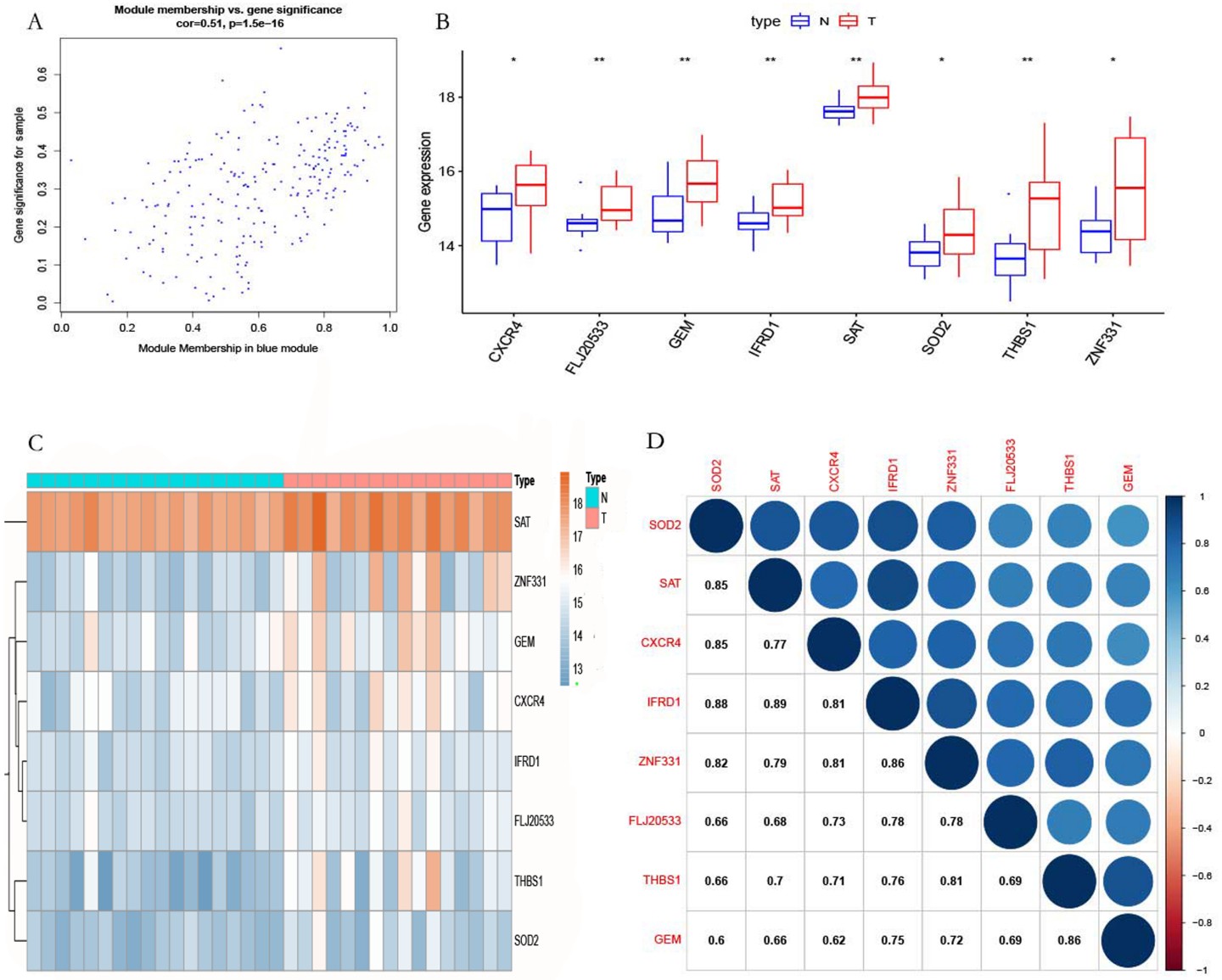

**Figure 7 Selected hub gene.** (A) The relationship between gene MM and GS. (B) Differential expression of the hub genes. (C) Heatmap of the hub genes in GSE12852. (D) Correlation analysis of the hub genes in GSE12852.

related coefficient between the infiltration of different immune cells and hub genes. The results reveal that activated mast cells and neutrophils had a positive correlation with the hub genes, while resting mast cells had a negative correlation with the hub genes (Table 1).

## DISCUSSION

Pelvic organ prolapse is caused by weakening of the supporting structures of the pelvic floor, resulting in the position of pelvic floor organs moving downward and the function becoming abnormal. POP seriously affects the quality of life of middle-aged and elderly women. In recent years, with the rapid development of gene sequencing and

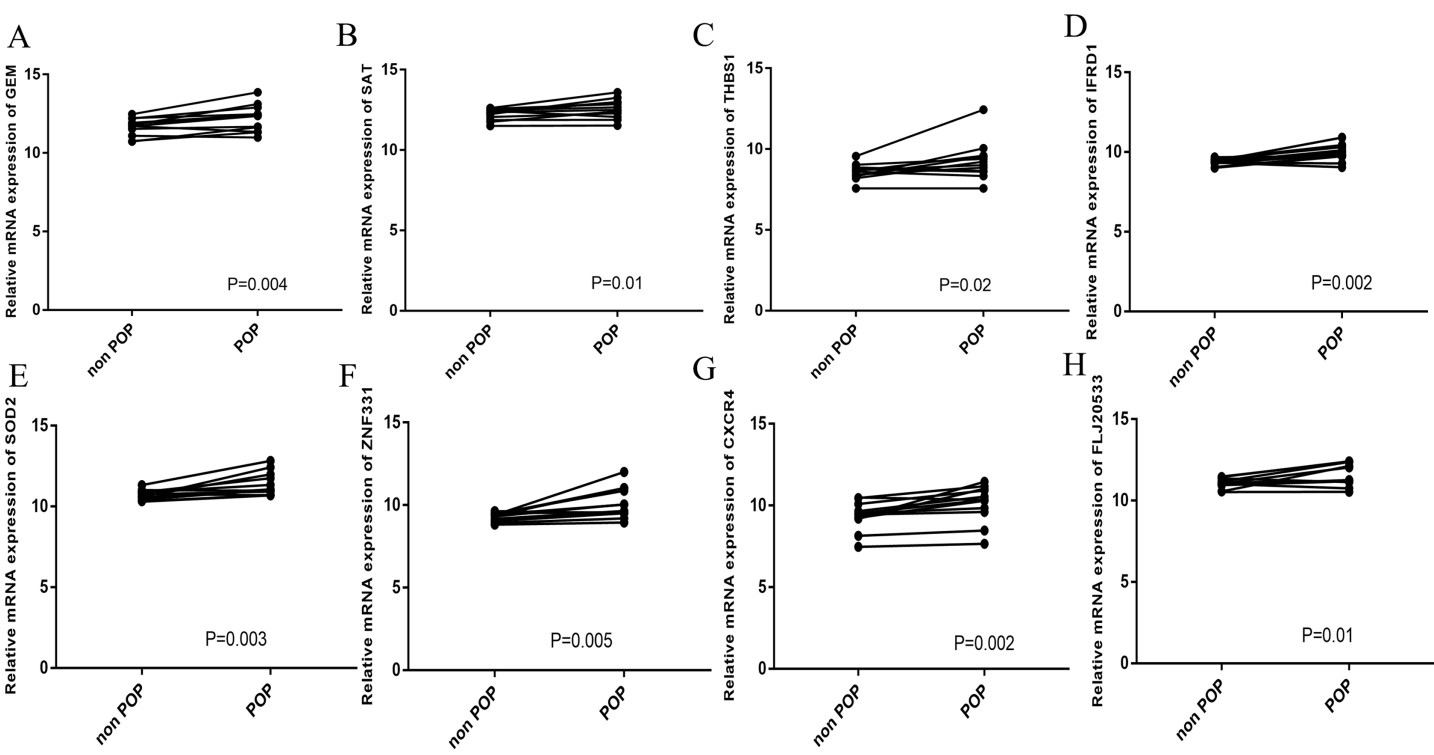

**Figure 8 Hub gene validation.** (A–H) RT-qPCR detection of the expression of hub genes in 12 pairs of POP and non-POP tissues.

bioinformatics technologies, further analysis and utilization of sequencing data have become possible. To explore the molecular mechanism related to the development of POP, eight hub genes (*ZNF331*, *THBS1*, *IFRD1*, *FLJ20533*, *CXCR4*, *GEM*, *SOD2*, and SAT) related to POP occurrence were screened out based on WGCNA and their biological functions were explored.

Thrombospondins (THBSs) are a group of glycoproteins that bind to collagen and tissue, participating in the interaction between cells and the extracellular matrix during the process of tissue development and repair. *THBS1* is the first member of the *THBS* gene family, which plays a significant role in many biological processes related to the occurrence and progression of cardiovascular diseases, such as angiogenesis, inflammation, and tissue remodeling (*Zhao, Isenberg & Popel, 2018*). In addition, *THBS1* can also affect tumor cell adhesion, invasion, migration, proliferation, apoptosis, and immune evasion (*Huang et al., 2017*). The overexpression of *THBS1* in POP tissues as compared with non-POP tissues has been predicted and verified (*Brizzolara, Killeen & Urschitz, 2009*), which is consistent with the results of our secondary analysis. *ZNF331* is located on chromosome 19q13, a recently cloned gene encoding a zinc-finger protein involved in thyroid tumorigenesis (*Babinger et al., 2007*). *ZNF331*, as a tumor suppressor gene, has also been reported to have low expression in colorectal cancer (*Wang et al., 2017*), esophageal cancer (*Jiang et al., 2015*), gastric cancer (*Yu et al., 2013*), and liver cancer (*Wang et al., 2013*), and its low expression is related to hypermethylation of its promoter region. However, the research on *ZNF331* in other diseases was rare. *Interferon-related*

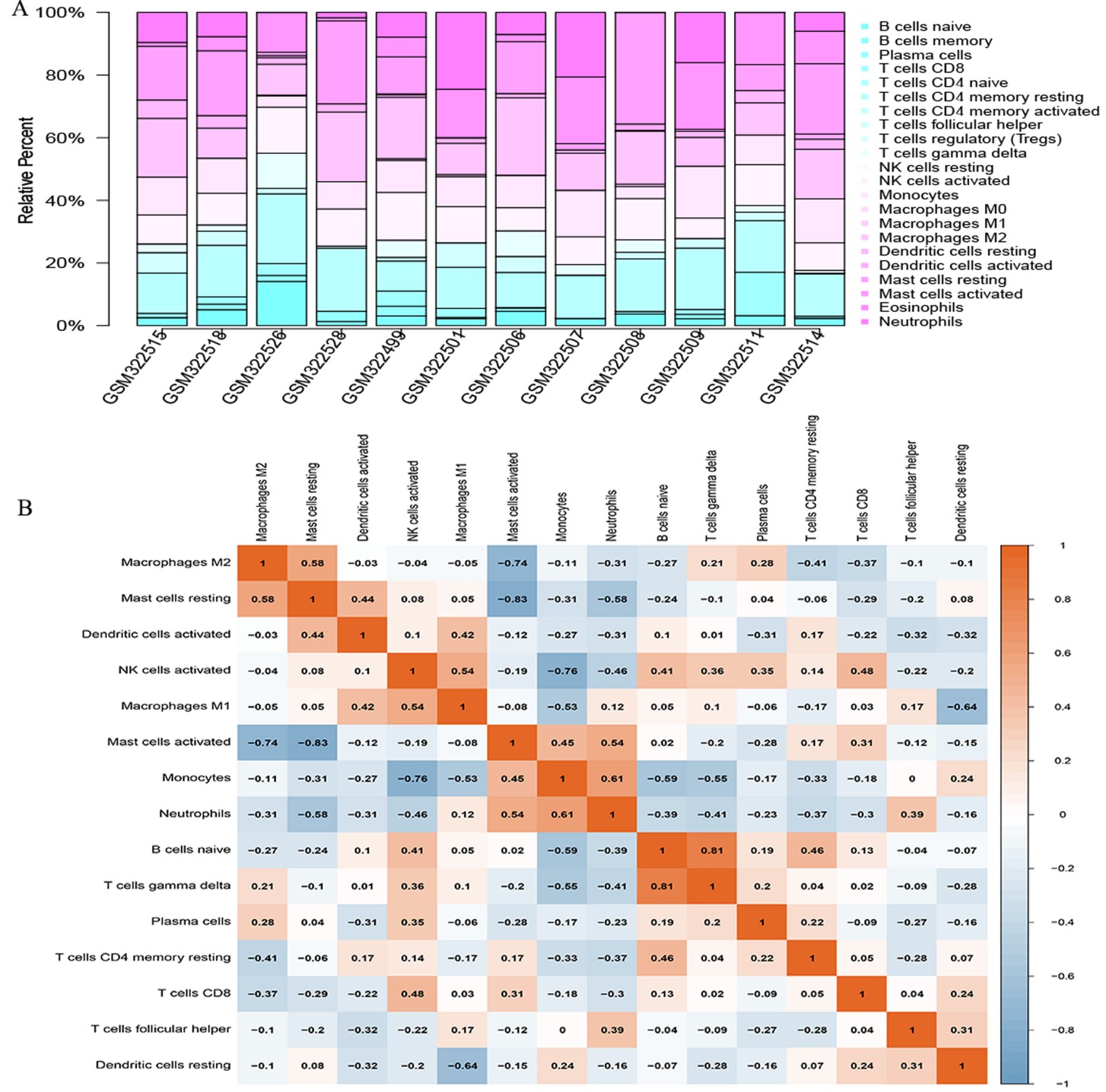

**Figure 9  The landscape of immune cell infiltration in GSE12852 (CIBERSORT *p* value < 0.05).** (A) Proportion of the 22 immune cell types in GSE12852. (B) Correlation matrix between the 22 immune cell types.

*developmental regulator 1* (*IFRD1*) is located on chromosome 7q22-q31 and acts a significant role in the development and differentiation of embryonic muscle cells (*Kraus, Haenig & Kispert, 2001*; *Lincoln, Alfieri & Yutzey, 2004*). *IFRD1* has been proved to be a

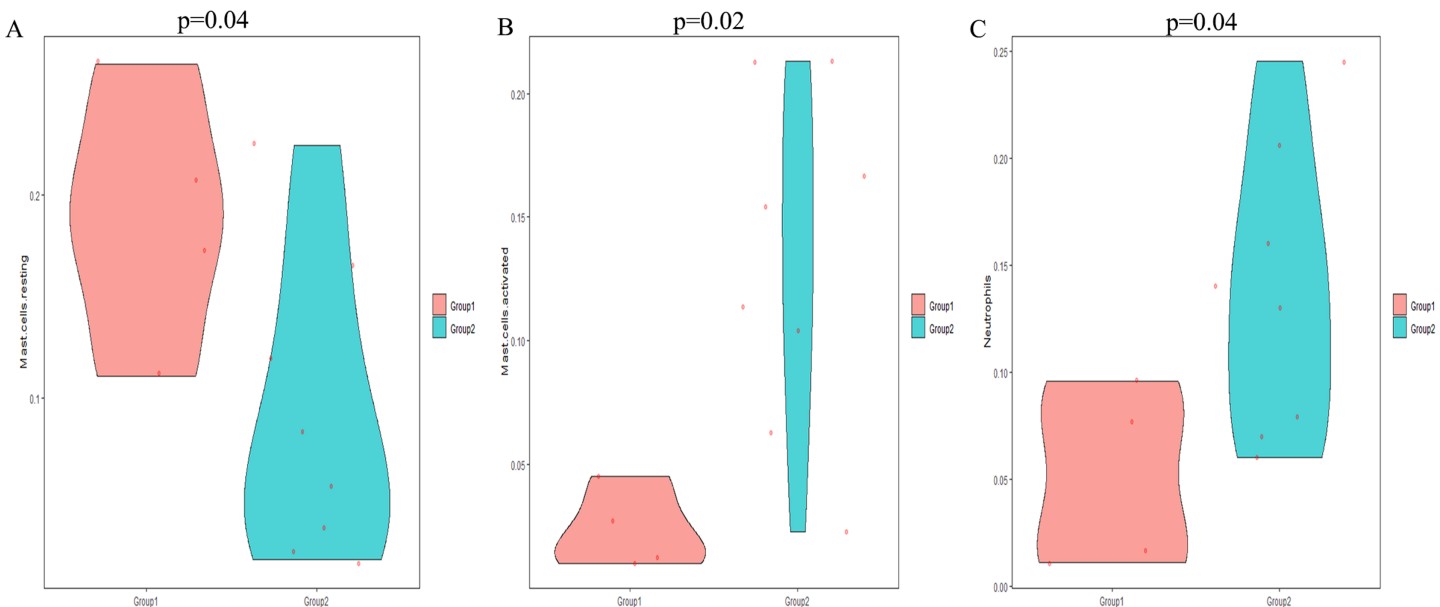

**Figure 10 The differential immune cell infiltration between POP and non-POP tissues.** (A) Resting mast cells; (B) activated mast cells; (C) neutrophils. Group 1: non-POP; Group 2: POP.

**Table 1 The correlation between hub genes and immune cell infiltration.**

| Gene | Immune cell | P | R |
|---|---|---|---|
| CXCR4 | Mast cells activated | 0.026025009 | 0.636596855 |
| FLJ20533 | Mast cells activated | 0.037410947 | 0.604312579 |
| GEM | Mast cells activated | 0.00382779 | 0.763874376 |
| IFRD1 | Mast cells activated | 0.010102021 | 0.707220887 |
| SAT | Mast cells activated | 0.026678483 | 0.634493564 |
| SOD2 | Mast cells activated | 0.019559382 | 0.659816924 |
| THBS1 | Mast cells activated | 0.023473376 | 0.645196019 |
| ZNF331 | Mast cells activated | 0.005806179 | 0.741209161 |
| GEM | Mast cells resting | 0.044390877 | −0.587895781 |
| IFRD1 | Mast cells resting | 0.010475671 | −0.704821204 |
| SAT | Mast cells resting | 0.029213165 | −0.626671202 |
| SOD2 | Mast cells resting | 0.002740399 | −0.780472841 |
| CXCR4 | Neutrophils | 0.002884121 | 0.778019124 |
| FLJ20533 | Neutrophils | 0.009179588 | 0.713441726 |
| GEM | Neutrophils | 0.030016479 | 0.624294555 |
| IFRD1 | Neutrophils | 0.001688177 | 0.802313109 |
| SAT | Neutrophils | 0.001387584 | 0.810472605 |
| SOD2 | Neutrophils | 2.12E−04 | 0.872716498 |
| THBS1 | Neutrophils | 0.014406502 | 0.682777143 |
| ZNF331 | Neutrophils | 0.002414909 | 0.786415881 |

modified gene of cystic fibrosis lung disease, which can regulate the effector function of neutrophil (*Gu et al., 2009*). Transmembrane protein 70 (TMEM70), also named FLJ20533, is a mitochondrial membrane protein that acts a role in the biosynthesis of mitochondrial ATPase (*Hejzlarova et al., 2011*). At present, research related to TMEM70 mainly focuses on cardiomyopathy and pulmonary hypertension. CXCR4, also known as CD184, is a highly conserved receptor for the chemokine CXCL12 (*Kashyap et al., 2017*). CXCR4 belongs to the *G* protein-coupled receptor superfamily and is expressed in a cortical protein-dependent manner on the cell surface (*Teicher & Fricker, 2010*). CXCR4 has been reported to be overexpressed in a variety of tumor cells and is involved in tumor proliferation, invasion, metastasis, and worse prognosis (*Ottaiano et al., 2020*; *Wang et al., 2020a*; *Zhu et al., 2020*). GTP-binding protein overexpressed in skeletal muscle (GEM) is a GTPase originally identified in mitogen-stimulated *T* lymphocytes and v-Abl-transformed pre-*B* cells and is highly expressed in the spleen, thymus, and kidneys (*Cohen et al., 1994*; *Huang et al., 2014*). Mitochondrial superoxide dismutase (SOD2) is an antioxidant enzyme that reduces the damage caused by oxidative stress to protect mitochondria (*Koltai et al., 2018*). Spermidine/spermine N1-acetyltransferase (SAT) is the rate-limiting enzyme in polyamine catabolism and functions through the acetylation of spermidine and spermidine to affect cell growth, proliferation, and apoptosis (*Pegg, 2008*, *2016*). Despite a detailed literature review, we found no reports of the involvement *ZNF331*, *IFRD1*, *FLJ20533*, *CXCR4*, *GEM*, *SOD2*, or *SAT* in POP. To explore the expression of the selected hub genes in POP and non-POP tissues, differential analysis of the GSE12852 dataset was performed by RT-qPCR. The results indicate that the selected hub genes were overexpressed in POP tissues as compared with non-POP tissues, suggesting that the selected hub genes may be related to POP occurrence.

GO and KEGG functional enrichment analyses were performed to study the function of the hub genes using the "R" software. GO functional annotation showed that the hub genes were mainly participated in the immune response. KEGG enrichment analysis revealed that the hub genes participated in the regulation of immune-related pathways such as the IL-17 signaling pathway. IL-17 is a member of the inflammatory cytokine family and is mainly produced by Th17 cells. IL-17 signaling has been reported to be associated with immunopathology and autoimmune diseases (*Amatya, Garg & Gaffen, 2017*). Previous research has confirmed that the extracellular matrix components in connective tissue can control the physical strength of the pelvic floor (*Dietz, Jarvis & Vancaillie, 2002*). The immune system can maintain the homeostasis of POP tissues by adjusting extracellular matrix components (*Yu et al., 2010*); thus, it is reasonable to suggest that regulation of the immune system is closely related to the occurrence of POP. At present, there is no direct evidence indicating that IL-17 is related to POP occurrence; however, we have reason to believe that the IL-17 signaling pathway plays a significant role in the development of POP given its role in immune regulation. In view of the close relationship between hub genes and immunity, we investigated immune cell infiltration in POP and non-POP tissues using the CIBERSORT algorithm. We found a higher infiltration of activated mast cells and neutrophils in POP tissues than in non-POP tissues,

while infiltration of resting mast cells was lower in POP tissues. Activated mast cells and neutrophils had a positive correlation with the hub genes, while resting mast cells had a negative correlation. According to the results, it is reasonable to suggest that the hub genes may be related to the development of POP by regulating the levels of activated/resting mast cells and neutrophils. Mast cells are important antigen-presenting cells that can release histamine and cytokines through degranulation and act a significant role in the occurrence and development of various inflammatory diseases (*Grabauskas et al., 2020*; *Novruzov, 2008*; *Sajay-Asbaghi et al., 2020*). Very recently, THBS1 was demonstrated to promote the inflammatory response of mast cells in chronic idiopathic urticaria and the permeability of human dermal microvascular endothelial cells by regulating the TGF-β/SMAD pathway, the effects of which can be inhibited by miR-194 (*Qu, Yang & Liu, 2020*). Several studies have shown that CXCR4 can promote mast cell chemotaxis to inflammatory sites (*Limón-Flores et al., 2009*; *Lv et al., 2019*; *Patadia et al., 2010*), and IFRD1 may involve in neutrophilic inflammation in cystic fibrosis (*Blanchard et al., 2011*; *Gu et al., 2009*; *Hector et al., 2013*). According to the current literature, there is no direct evidence to prove the accuracy of our prediction results; however, the relationship between the hub genes and infiltration of different immune cells suggests they are correct.

Some limitations of this research need to be discussed. Firstly, the research remained at the prediction stage and there is insufficient experimental evidence to verify our prediction results. In the future, verification in vitro and in vivo should strengthen our observations. Secondly, the sample size for this study was not large enough due to the limited datasets in the database, which may have caused some bias. Thirdly, the samples of our research were coming from the ligamentum. The ligamentum are full of fibroblasts and it is not sure that they capture many white blood cells. We could not guarantee the accuracy of the CIBERSORT algorithms completely due to the blood specificity of the algorithms.

## CONCLUSIONS

Our research identified eight hub genes and three immune cell types that may be related to POP occurrence. These hub genes may participate in the pathogenesis of POP by regulating the immune environment, giving them certain diagnostic and therapeutic value in POP.

## ACKNOWLEDGEMENTS

We thank the authors who provided the GEO public datasets.

### Funding

This article was funded by Shengjing Hospital of China Medical University. The funders had no role in study design, data collection and analysis, decision to publish, or preparation of the manuscript.

## Grant Disclosures

The following grant information was disclosed by the authors:
Shengjing Hospital of China Medical University.

## Competing Interests

The authors declare that they have no competing interests.

## Author Contributions

- Ying Zhao conceived and designed the experiments, performed the experiments, analyzed the data, prepared figures and/or tables, authored or reviewed drafts of the paper, and approved the final draft.
- Zhijun Xia conceived and designed the experiments, analyzed the data, prepared figures and/or tables, authored or reviewed drafts of the paper, and approved the final draft.
- Te Lin performed the experiments, analyzed the data, authored or reviewed drafts of the paper, and approved the final draft.
- Yitong Yin performed the experiments, analyzed the data, prepared figures and/or tables, authored or reviewed drafts of the paper, and approved the final draft.

## Human Ethics

The following information was supplied relating to ethical approvals (i.e., approving body and any reference numbers):

The Ethics Committee of Shengjing Hospital of China Medical University approved the study protocol in accordance with the guidelines of the Declaration of Helsinki (No. 2018Ps68K), in accordance with the guidelines of the Helsinki Declaration.

## Data Availability

The sequence is available at NCBI GEO (GSE12852) and the PCR data are available as a Supplemental File.

## Supplemental Information

Supplemental information for this article can be found online at http://dx.doi.org/10.7717/peerj.9773#supplemental-information.

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
