# Peer review of "Significance of hub genes and immune cell infiltration identified by bioinformatics analysis in pelvic organ prolapse"

_PeerJ, doi:10.7717/peerj.9773_

## Round 0.1 · original submission · Major Revisions

The manuscript needs a complete rewrite according to both reviewers' comments. It misses the exact parameters and the reader can be lost when how many samples are used. Also, reasoning regarding the selection criteria of this dataset is needed. The parameters usage of STRING is not clear, as well. The manuscript's English needs improvement. However, I think the manuscript has merits and the usage of WGCNA and CIBERSORT algorithm can show novel insights in pelvic organ prolapse.

·

Basic reporting

Zhao et al analyzed a published transcriptomic data set from GEO to select the most important genes (hub genes) having a possible essential role in Pelvic organ prolapse (POP), besides the authors discovered the immune cells infiltration and significant changes have been found during the disease compared to the healthy state. Not only Weighted Gene Co-expression Network Analysis (WGCNA) has been used to find the hub genes in the network but also RT-qPCR helped to validate the results.
The authors collected the most important articles in the topic, they published their work mostly in an understandable way. However, the language was not accurate and scientific enough in every case (and should be rewrite at some parts), this did not disrupt their work.
The figures and tables are numbered good, however the sequence of the figures is not satisfied. The quality of them is adequate, but I missed a general figure about the workflow which could help the interpretation of the work. Another general problem, that at some points the description is not detailed enough, it is difficult to understand the method, after rereading the section neither.

Experimental design

The research question has been well defined, looking for hub genes / potential biomarkers in POP is an essential step to identify and treat the early stage of the disease successfully. The authors collected the most relevant publications in this research area. The main gap has been identified and the written method tried to answer it. The bioinformatic analysis used various areas of biology (hierarchical clustering, WGCNA, GO and pathway analysis, CIBERSORT etc.), however not every step was understandable enough due to the lack of documentation (e.g. parameter description), therefore I suggest to expand the text with the provided information in the pdf.

Validity of the findings

The authors published a well-built pipeline to collect potential biomarkers from the published data set. Nonetheless, I have concerns about the unbalance regading the number of samples in different conditions (at the beginning it was 18 vs 16 but it could not be followed after removing the outlier samples), while by the CIBERSORT analysis these numbers became 4 vs 8.
In spite of that the workflow may contain mistakes, by the discussion part the authors explained that the found 8 genes could be highly related to the POP supported by the literature.

Additional comments

I would like to describe my comments in the PDF document.

Reviewer 2 ·

Basic reporting

The writing Language need to be improved by professionals.
background, reference, and study design are meet the academic level.

Experimental design

the experimental design is insufficient.
Only one dataset involved in this study, which contained only 34 samples. The dataset is too small, more dataset is need to show the reliable of results.
And the authors need provide the search strategy, which means how this dataset was chose (including database, inclusion criteria, exclusion criteria, key words).
Which and how these analysis software used, the authors need described briefly.

Validity of the findings

The quality of sample in GSE12852 needs to be analyzed and described.
How the normal sample of GSE12852 used in data analysis, which need to be provided this study?
The authors described some RNA were extracted from POP samples? Where are the POP tissues from? Human? Animal? There is no information about how to collect, where to collect the how the sample process. This part is very confused.

Additional comments

Some contents in the article is very confused, and cannot be understand. For example, line 45 “the incidence of the POP will be as high as 46% in 2050(Wu et al. 2009)”. Or line 59 ” complex network theory of high-throughput data emerges as the Times require”
Some details need to be provided when described for the 1st time, such as in the results part of abstract” WGCNA revealed that blue module”. The authors need to describe the blue module

---

## Round 0.2 · Minor Revisions

The manuscript improved significantly. I have to agree with the reviewer about her points

Please add the exact parameters of the WGCNA method and elaborate on the selection of the most significant module. The background is missing still on the DAVID analysis. If it is the whole human genome it is possibly biased, due to not the whole human genome is on microarrays. Please as well mention the limitation of the study which is coming from the ligamentum samples and the CIBERSORT algorithms blood specificity. The samples are full of fibroblasts and it is not sure that they capture many white blood cells.

·

Basic reporting

Zhao et al analyzed a published transcriptomic data and created a Weighted Gene Co-expression Network Analysis (WGCA) to identify hub genes which can have an effect on Pelvic organ prolapse. The updated version of the manuscript describes very well not only the disease itself with current statistics but the method part became much more understandable. The language is improved, I have found only a few typos/mistakes. The new workflow figure is appreciated, it helps to follow the analysis. The network figure (Figure 5) should be improved because the message of the network can not be seen (neither the name of the nodes). I would suggest to avoid the usage of the simple circle layout, rather grouping the proteins with high degree and arrange their connections around them. Figure 2 has a small mistake in the description: menopause state is assigned with 1 not with 0!

Experimental design

The research question has been well defined, the methodology is well-described. However, the definition of the final beta value ( I assume it is 7 based on the figures but it would be great to describe) is missing from the text and the hub gene module identification paragraph is a bit confusing (why ME and not MS has been used for the selection of hub module). In the functional enrichment, the background dataset is still not good. Finally, the limitation of CIBERSORT algorithm using the described samples was not mentioned in the text.

Validity of the findings

The outcome of the whole pipeline was a list of 8 hub genes which were validated by RT-qPCR. With the extended description of methods/parameters, I do not have any concern regarding the results.

Additional comments

PDF is attached containing the comments.

---

## Round 0.3 · Minor Revisions

I have checked the manuscript. I have a few questions because I have not seen that any change was made regarding the previous revision:

1. Figure 6 has not changed compared to the previous revision. Meanwhile, it is stated that the background of the analysis was changed for the used chip. Why?

2. What is the reason for selecting S100A9 and IL6 in
Figure 5?

Otherwise, the manuscript is ready to be published.

---

## Round 0.4 · accepted · Accept

Thank you for the quick response. So the same background was used all the time for the analysis it was just not mentioned originally.